# SARS-CoV-2 infection remodels the host protein thermal stability landscape

Joel Selkrig[1,†] (iD), Megan Stanifer[2,†] (iD), André Mateus[1,†] (iD), Karin Mitosch[1,†], Inigo Barrio-Hernandez[3,†], Mandy Rettel[4,†], Heeyoung Kim[2], Carlos G P Voogdt[1], Philipp Walch[1,5], Carmon Kee[2] (iD), Nils Kurzawa[1,5] (iD), Frank Stein[4] (iD), Clément Potel[1], Anna Jarzab[6], Bernhard Kuster[6] (iD), Ralf Bartenschlager[2,7,8] (iD), Steeve Boulant[9,10,*] (iD), Pedro Beltrao[3,**] (iD), Athanasios Typas[1,***] (iD) & Mikhail M Savitski[1,****] (iD)

## Abstract

The severe acute respiratory syndrome coronavirus 2 (SARS-CoV-2) is a global threat to human health and has compromised economic stability. In addition to the development of an effective vaccine, it is imperative to understand how SARS-CoV-2 hijacks host cellular machineries on a system-wide scale so that potential host-directed therapies can be developed. *In situ* proteome-wide abundance and thermal stability measurements using thermal proteome profiling (TPP) can inform on global changes in protein activity. Here we adapted TPP to high biosafety conditions amenable to SARS-CoV-2 handling. We discovered pronounced temporal alterations in host protein thermostability during infection, which converged on cellular processes including cell cycle, microtubule and RNA splicing regulation. Pharmacological inhibition of host proteins displaying altered thermal stability or abundance during infection suppressed SARS-CoV-2 replication. Overall, this work serves as a framework for expanding TPP workflows to globally important human pathogens that require high biosafety containment and provides deeper resolution into the molecular changes induced by SARS-CoV-2 infection.

**Keywords** aryl hydrocarbon hydroxylase; heat shock chaperone; rhapontigenin; SARS-CoV-2; tanespimycin

**Subject Categories** Microbiology, Virology & Host Pathogen Interaction; Proteomics

Mol Syst Biol. (2021) 17: e10188

## Introduction

SARS-CoV-2, the novel severe acute respiratory syndrome coronavirus, has led to a worldwide pandemic that is upending economic stability and poses a tremendous burden on healthcare systems. As the pandemic is ongoing and treatment options remain limited, a deeper molecular understanding of the host proteins modulated by SARS-CoV-2 to promote its infection life cycle is urgently required to expose potential therapeutic intervention strategies. SARS-CoV-2 hijacks host cell processes for its replication, packaging and release, leading to alterations in cell signalling and protein expression (Bojkova *et al*, 2020; Bouhaddou *et al*, 2020; Zecha *et al*, 2020; Gordon *et al*, 2020b). Although genetic approaches are shedding light onto the functional relevance of such large-scale proteomic information (preprint: Wang *et al*, 2020; Wei *et al*, 2020; Gordon *et al*, 2020a), complementary approaches are needed to expand our understanding of the biophysical changes that proteins can undergo within a live infection context.

Many physiological changes to the functional state of a protein are reflected in altered protein thermal stability (Mateus *et al*, 2016, 2020b). Thermal proteome profiling (TPP) measures protein thermal stability *in situ* on a proteome-wide scale and is a powerful tool for identifying proteins with altered biophysical states in living cells: protein–protein interactions (PPIs), the reorganization

1 Genome Biology Unit, European Molecular Biology Laboratory (EMBL), Heidelberg, Germany
2 Department of Infectious Diseases, Molecular Virology, Heidelberg University Hospital, Heidelberg, Germany
3 European Bioinformatics Institute (EMBL-EBI), Hinxton, UK
4 Proteomics Core Facility, European Molecular Biology Laboratory (EMBL), Heidelberg, Germany
5 Faculty of Biosciences, EMBL and Heidelberg University, Heidelberg, Germany
6 Proteomics and Bioanalytics, Technical University of Munich, Freising, Germany
7 Division "Virus-associated Carcinogenesis", German Cancer Research Center (DKFZ), Heidelberg, Germany
8 German Center for Infection Research, Heidelberg Partner site, Heidelberg, Germany
9 Department of Infectious Diseases, Virology, Heidelberg University Hospital, Heidelberg, Germany
10 Research Group "Cellular Polarity and Viral Infection", German Cancer Research Center (DKFZ), Heidelberg, Germany
*Corresponding author. Tel: +49 6221567865; E-mail: s.boulant@dkfz-heidelberg.de
**Corresponding author. Tel: +44 1223494610; E-mail: pbeltrao@ebi.ac.uk
***Corresponding author. Tel: +49 62213878156; E-mail: typas@embl.de
***Corresponding author. Tel: +49 62213878560; E-mail: mikhail.savitski@embl.de
†These authors contributed equally to this work

of protein complexes (Mateus *et al*, 2018; Becher *et al*, 2018), post-translational modifications (Huang *et al*, 2019; preprint: Potel *et al*, 2020) or the interaction of proteins with co-factors, small molecules or nucleic acids (Sridharan *et al*, 2019; Mateus *et al*, 2020a). The method quantifies protein abundance changes in the same samples, offering a comprehensive insight into the physiological changes that occur during cellular perturbations. Applying TPP to viral pathogens can provide a comprehensive and orthogonal view on various protein state changes during infection and highlight key proteins and cellular processes required for viral replication (Hashimoto *et al*, 2020). However, no study to date has applied TPP to the infection context under biosafety level 3 (BSL3) conditions.

Here, we present the first dataset on the time-resolved interrogation of host protein thermostability changes during infection of Caco-2 cells with SARS-CoV-2. Hundreds of proteins changed in abundance and thermal stability, starting from as early as 1 h postinfection (hpi), indicating rapid remodelling of the host cell proteomic state. Integration of our data with SARS-CoV-2 induced phosphorylation events and viral–host PPI maps revealed common cellular processes, such as RNA splicing, that are affected within the first 1–2 h postinfection, followed by disturbances in cell cycle and the cell cytoskeleton towards later stages of infection. Additional proteins and processes not previously implicated in SARS-CoV-2 infection were also affected. To illustrate that changes in thermal stability and abundance—as a result of SARS-CoV-2 infection—can be used to unravel host cell biology that the virus hijacks or depends on, we tested pharmacological inhibitors for a subset of affected proteins to modulate viral proliferation. This led us to uncover a requirement for host heat shock chaperones (HSP90) and the aryl hydrocarbon hydroxylase (CYP1A1) in supporting SARS-CoV-2 proliferation. In conclusion, we report an important method development for the TPP workflow that expands its application to high biosafety conditions, thus paving the way for future studies aimed at understanding how dangerous human pathogens remodel the host proteome state. This allowed us to implicate new host proteins and cellular pathways in SARS-CoV-2 infection that can be pharmacologically targeted to inhibit viral replication.

## Results

### Proteome abundance and thermal stability upon SARS-CoV-2 infection

Since viral pathogens need to hijack endogenous host protein machinery for their replication, we reasoned that TPP would be a powerful means to uncover functionally relevant changes during infection with SARS-CoV-2 (Savitski *et al*, 2014). To generate proteome-wide thermal stability profiles in a typical TPP workflow, cell aliquots are subjected to ten different temperatures to promote *in situ* protein unfolding. Cells are then lysed, insoluble proteins are removed, and the remaining soluble protein fraction at each temperature is collected and analysed with mass spectrometry-based quantitative proteomics (Becher *et al*, 2016, 2018). To ensure compatibility with a BSL3 working environment, we first adapted our standard TPP protocol (Franken *et al*, 2015) by

replacing centrifugation steps—which can otherwise lead to the generation of aerosols containing airborne pathogens—with a filtration step aided by vacuum for protein aggregate removal that can be performed inside a HEPA filtered laminar flow cabinet (Fig 1A; see Materials and Methods for details). We confirmed that this step was as effective as the previous centrifugation procedure (Appendix Fig S1A and B).

We then applied the adapted TPP workflow to study how SARS-CoV-2 infection modifies host proteome thermal stability and abundance. Caco-2 cells, which are permissive to SARS-CoV-2 infection (Stanifer *et al*, 2020; Bojkova *et al*, 2020), were infected in triplicate with SARS-CoV-2 at a multiplicity of infection (MOI) of 0.5 for 1 h, after which unbound viral particles were removed and samples were harvested at 1, 2, 4, 7, 12, 24 and 48 h postinfection (Fig 1A). Four hours after addition of the virus ~ 5.2% of cells were infected, with cell infection rates then peaking at around 18% by 24 h (Fig 1B and C). At each time point, samples were harvested and processed using vacuum-based aggregate removal as described above and analysed by mass spectrometry-based quantitative proteomics (Fig 1A).

We detected 7,414 proteins with at least two unique peptides. This included three viral proteins: the spike glycoprotein (S), the nucleoprotein (N) and Orf9b. Abundance changes of these three proteins over time largely recapitulated the microscopy observations for viral replication (Fig 1D). SARS-CoV-2 proliferation kinetics and the viral protein coverage were comparable to those reported previously in Caco-2 cells (Bojkova *et al*, 2020). We then calculated abundance changes relative to the non-infected control for 5,564 proteins (detected at the two lowest temperatures in at least two replicates), and thermal stability changes for 4,076 proteins (detected at the two lowest temperatures in at least two replicates and overall detected in at least ten temperatures; Dataset EV1; see Materials and Methods for details). Of these, 350 proteins showed significant changes in abundance ($|z\text{-score}| > 1.96$ and $q$-value $< 0.05$; see Materials and Methods for details) and 278 proteins showed changes in thermal stability in at least one time point (Dataset EV1, Appendix Fig S2). Consistent with previous observations, the majority of protein abundance changes were the result of down-regulation (62.9% of overall abundance changes; 220 proteins in total) (Bouhaddou *et al*, 2020), whereas stability changes were dominated by protein destabilization (61.9% of overall thermal stability changes: 172 proteins in total) (Fig 2A and Appendix Fig S2). For instance, we observed an early increase in the abundance of host transcription and translation-related proteins (including viral-associated processes), specifically within the first hour of infection, whereas thermostability changes for these processes were mostly sustained throughout the infection time course (Fig 2B and Dataset EV2). Proteins associated with mitochondrial translation changed only in abundance within the first 2 h of infection, and proteins related to cytoskeletal remodelling and protein folding changed only in thermal stability mostly towards the later stages of infection (Fig 2B and Dataset EV2). Hits related to protein folding were mostly the result of thermal destabilization and cell–cell adhesion proteins were mostly thermally stabilized (Appendix Figs S3 and S4, and Dataset EV2). In summary, we reveal a rich and dynamic map of host protein abundance and thermal stability changes upon SARS-CoV-2 infection that may represent novel therapeutically relevant targets.

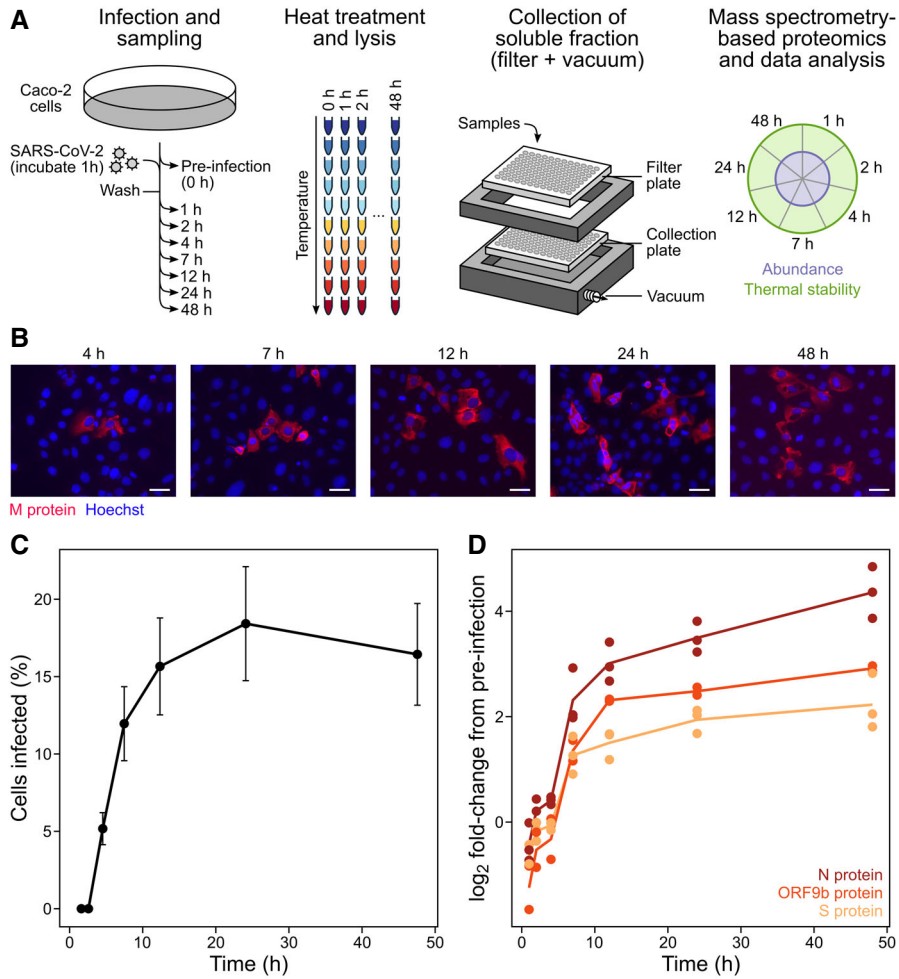

**Figure 1. Global assessment of host abundance and thermostability changes during SARS-CoV-2 infection.**

A   Caco-2 monolayers consisting of 1.5e$^6$ cells per sample were infected with SARS-CoV-2 at an MOI of 0.5 (see Materials and Methods). Three biologically independent replicates were collected per time point. Samples were subjected to vacuum manifold-based 2D-TPP. After removal of insoluble proteins, the remaining soluble fractions were labelled with isobaric mass tags (TMTpro) to enable protein quantification. Samples were combined in a 2D-TPP layout in order to compare protein thermal stability and abundance throughout the infection time course (Becher *et al*, 2018). The uninfected ($t_{-1}$) time point was used as a reference to calculate fold changes (FC). Significant protein changes (abundance and thermal stability) are represented as circle plots for individual proteins, as previously described (Becher *et al*, 2018). Inner circle corresponds to protein abundance and the outer circle corresponds to thermal stability changes ($n = 3$).

B   Caco-2 cells were infected with SARS-CoV-2 as per (A). Cells were washed, fixed and stained with DAPI (blue) to visualize cell nuclei and incubated with anti-N-protein antibody (red) to visualize SARS-CoV-2 infected cells. Scale bar denotes 10 μm.

C   Infection rates were calculated using automated imaging software (see Materials and Methods) calculated from six fields of view per time point per biological replicate. Data point indicates the sample means and the error bars the standard error of the mean (SEM) ($n = 3$).

D   SARS-CoV-2 detected proteins plotted as a function of time where each data point denotes a different replicate ($n = 3$).

## SARS-CoV-2-induced thermal stability changes converge on viral–host PPIs and phosphosite regulation

Next, we asked whether proteins that change in abundance or thermal stability during SARS-CoV-2 infection also participate in viral–host PPIs and/or possess phosphosites that are differentially regulated upon SARS-CoV-2 infection (Bouhaddou *et al*, 2020; preprint: Samavarchi-Tehrani *et al*, 2020; Gordon *et al*, 2020b). To do this, we compared this list of proteins to previously acquired proteome-wide datasets describing SARS-CoV-2-host PPIs and phosphosite regulation upon infection (Bouhaddou *et al*, 2020; Gordon *et al*, 2020b). We found an overlap of 30 proteins that had altered abundance or

thermal stability and were previously shown to physically interact with viral baits in affinity purification mass spectrometry (AP-MS) experiments (Gordon *et al*, 2020b; Appendix Fig S5A) and 204 proteins in BioID experiments (preprint: Samavarchi-Tehrani *et al*, 2020) (Appendix Fig S5B). Additionally, 107 of the proteins with altered abundance or thermal stability also exhibited altered phosphorylation states during SARS-CoV-2 infection (Appendix Fig S5C; Bouhaddou *et al*, 2020). Thus, proteins exhibiting multiple alterations in abundance, thermal stability and/or their interaction or phosphorylation state provide further molecular support for their functional role during SARS-CoV-2 infection. Importantly, the majority of proteins changing in thermal stability were unique to the TPP

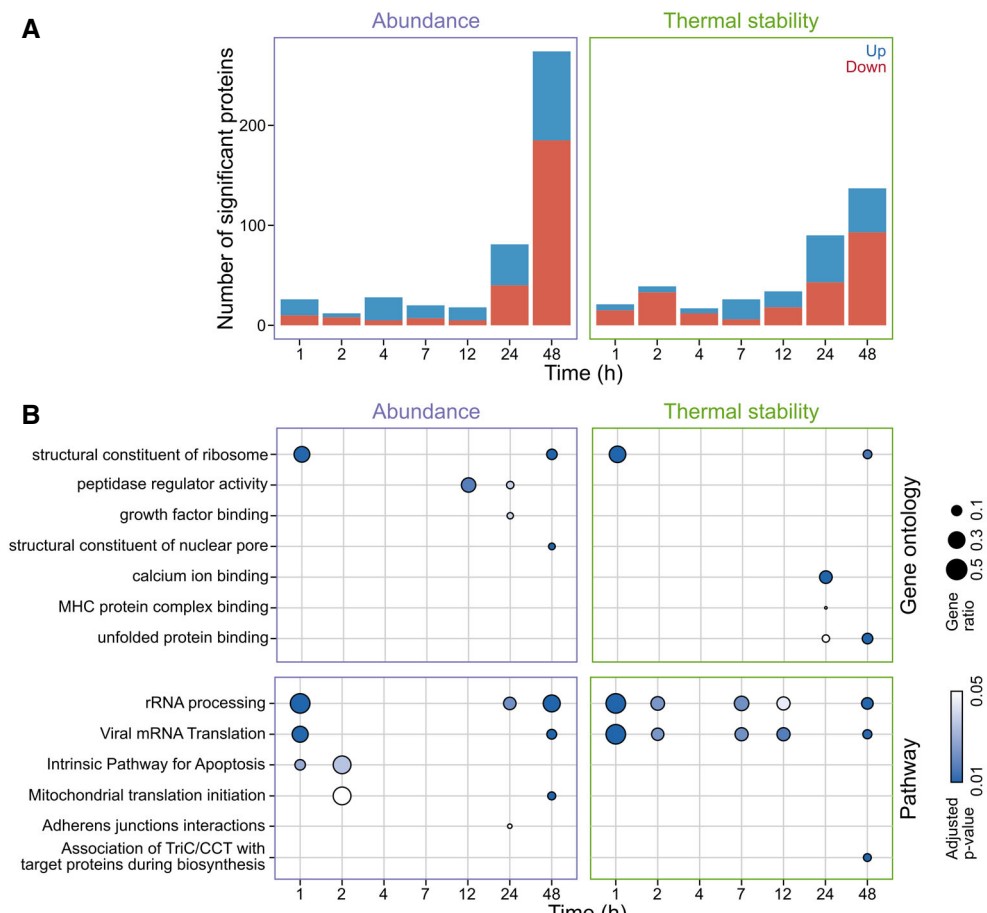

**Figure 2. Global protein abundance and thermal stability changes upon SARS-CoV-2 infection.**

A    Total number of significantly up- or down-regulated proteins (|z-score| > 1.96, q-value < 0.05) per time point of SARS-CoV-2-infected Caco-2 cells as described in Fig 1A. Proteins that were significantly changed in abundance (left) and thermal stability (right) are sub-grouped according to either down-regulation/destabilization (red) or up-regulation/stabilization (turquoise).

B    Gene Ontology (GO) biological process and pathway enrichment of selected pathways. See Dataset EV2 for complete GO term biological process and pathway enrichments.

experiment. As discussed below, some of these hits are direct targets of SARS-CoV-2 inhibiting drugs, suggesting that TPP provides orthogonal information on the biology of SARS-CoV-2 infection.

To understand how thermal stability changes relate to alterations in host cellular processes, we used proteins with altered thermal stability to seed a network which was then propagated using host proteins previously shown to interact with SARS-CoV-2 proteins or

exhibit disturbances in phosphosite regulation during SARS-CoV-2 infection (Bouhaddou *et al*, 2020; Gordon *et al*, 2020b; Fig 3A). After network propagation, clustering was performed to split the network into modules containing proteins with related biological functions. Biological pathway analysis revealed pronounced convergence of host proteome regulation (thermal stability or phosphosite) and physical interactions (viral–host PPIs) within core cellular

**Figure 3. Network analysis of integrated SARS-CoV-2 datasets.**

A    Network of proteins commonly altered or targeted during SARS-CoV-2 infection. Proteins displaying significantly altered changes in thermal stability were used as seeds for building the network. SARS-CoV-2 infection regulated phosphosites and their predicted upstream kinases as well as protein–protein interactions between viral baits and host prey proteins were used to propagate the network. After network propagation, clustering was performed to split the network into modules (see Materials and Methods). GO term enrichments for each module are shown bottom right. Protein thermal stability data is from this study, whereas phosphorylation/kinase regulation (Bouhaddou *et al*, 2020) and viral–human protein–protein interaction (Gordon *et al*, 2020b) data were acquired from previous work and used to propagate the TPP hits that seed this network. All TPP thermal stability hits per module are displayed.

B    Two statistical tests were then performed on the generated network from (A). First, to assess whether the modules are enriched in TPP hits (Fisher test), and second, to determine whether they received a significant amount of starting signal (KS test). Dashed red line represents significance threshold (P-adj < 0.05).

C    Heat map showing the time point at which TPP hits are found within each module defined in (A). Green tile indicates significant changes in thermal stability.

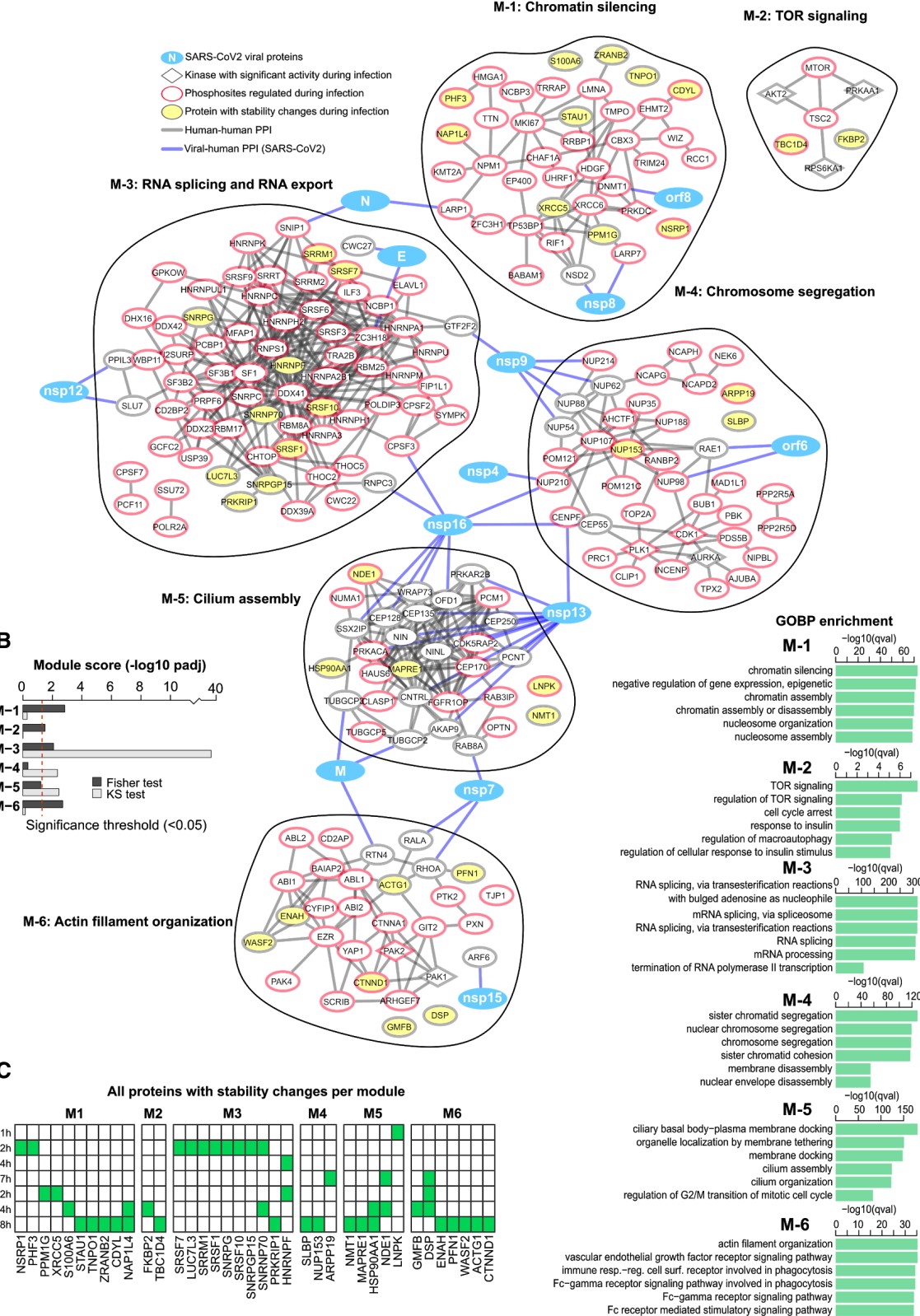

**Figure 3.**

processes including proteins involved in cell cycle regulation, microtubule organization and mRNA splicing regulation (Fig 3A and B). Notably, proteins involved in RNA splicing (Module M-3) (e.g. SNRNP70, SNRPGP15, SRSF10, SNRPG, SRSF1, SRSF7, SRRM1 and LUC7L3) were consistently thermally destabilized at 2 h postinfection (Fig 3C). Disturbances in the RNA splicing machinery is in line with recent reports demonstrating alterations in the abundance (Bojkova et al, 2020) and phosphorylation states (Bouhaddou et al, 2020) of RNA splicing-related proteins during SARS-CoV-2 infection. Importantly, the functional requirement for spliceosome function in SARS-CoV-2 infection has been previously demonstrated using the spliceosome inhibitor, pladienolide B, which suppressed SARS-CoV-2 replication (Bojkova et al, 2020). Interestingly, the SARS-CoV-2 protein nsp16 disrupts mRNA splicing events through binding of the mRNA recognition domains U1/U2 snRNAs, thus preventing translation of antiviral gene products (Boudreault et al, 2019; Maranon et al, 2020; Banerjee et al, 2020).

Furthermore, we observed pronounced changes in the thermal stability of structural components of desmosomes which are involved in cell–cell adhesion; desmoplakin (DSP) was thermally destabilized between 7 and 24 h of infection while desmoglein-2 (DSG2) and plakophilin-2 (PKP2) showed thermal stabilization towards later time points (Fig 3A, and Dataset EV1). This may have implications for the spread of SARS-CoV-2, as other viruses have been shown to target cell–cell junction proteins to facilitate their spread to neighbouring cells (Mateo et al, 2015).

We detected a strong destabilization of inosine-5′-monophosphate dehydrogenase 2 (IMPDH2) starting from 4 h postinfection (Fig 4A). IMPDH2 catalyses the conversion of inosine 5′-phosphate (IMP) to xanthosine 5′-phosphate (XMP), which is a rate limiting step of de novo guanine biosynthesis. Inhibition of IMPDH2 with the antiviral ribavirin was previously shown to suppress SARS-CoV-2 replication (Bojkova et al, 2020), as does its genetic ablation (Gordon et al, 2020a). These results provide further evidence that IMPDH2 is functionally modified during infection, possibly by a direct interaction with nsp14, and provides additional molecular insight into why IMPDH2 pharmacological inhibition effectively suppresses SARS-CoV-2 replication.

We found that the glycosaminoglycan biosynthesis protein UDP-glucose 6-dehydrogenase, UGDH and the proteasomal 26S inhibitor, PAAF1, were both destabilized upon SARS-CoV-2 infection (Fig 4B). Importantly, both UGDH and PAAF1 are required for SARS-CoV-2 infection (preprint: Wang et al, 2020), and nsp6 was previously found to physically interact with UGDH, and the viral membrane glycoprotein M with UGDH and PAAF1 in BioID experiments (preprint: Samavarchi-Tehrani et al, 2020) (Fig 4B). These data suggest that thermal destabilization of UGDH and PAAF1 during SARS-CoV-2 infection capture functionally relevant changes in their biophysical state that impact SARS-CoV-2 proliferation.

We previously reported that protein phosphorylation, in some cases, corresponds to alterations in protein thermal stability (preprint: Potel et al, 2020). We noticed that some phosphorylation events we previously observed (Bouhaddou et al, 2020) during SARS-CoV-2 infection temporally coincided with protein thermal stability changes, particularly during earlier time points. For instance, phosphorylation of S604 on the transcription factor scaffold attachment factor B1 (SAFB), temporally coincided with its destabilization which began at 1 hpi and reached significance at 2 hpi

(Bouhaddou et al, 2020; Fig 4C). We have previously demonstrated that phosphorylation of S604 is associated with SAFB thermal destabilization (preprint: Potel et al, 2020; Fig 4D), suggesting SARS-CoV-2 infection promotes SAFB thermal destabilization by phosphorylation on S604. Consistent with a role for S604 in regulating SAFB function, the protein abundance of SAFB target genes FBL, RPS15 and TAF15 increased concomitantly with S604 levels (Appendix Fig S6A). Notably, the SAFB target gene product LARP1 was previously found to interact with the SARS-CoV-2 N-protein, and the target gene neuroguidin (NGDN) with the SARS-CoV-1 protein nsp9 (Gordon et al, 2020a, 2020b). Confirming the functional relevance for SARS-CoV-2, NGDN was recently shown to be required for SARS-CoV-2 proliferation (preprint: Wang et al, 2020). These findings allude to a potential temporal interplay between SARS-CoV-2 and SAFB, whereby phosphorylation of S604 on SAFB that induces the expression of proteins directly targeted by SARS-CoV-2 during infection.

Several host chaperones had profound changes in thermal stability during SARS-CoV-2 infection. For instance, the core components of the human chaperone complex (TRiC/CCT) were mildly thermally stabilized early in infection, whereas they were strongly destabilized at later infection stages (Appendix Fig S6B). This complex is required for influenza virus replication, where it was shown to form a protein complex with the viral RNA polymerase subunit PB2 (Fislová et al, 2010). In addition, several host heat shock chaperones were thermally destabilized (e.g. HSP90AB1, HSP90AA1, HSPB1, HSPA8), some of which also contain differentially regulated phosphosites upon SARS-CoV-2 infection (Bouhaddou et al, 2020). For example, thermal destabilization of HSPB1 was associated with increased phosphorylation of S15, S78 and/or S82 (Appendix Fig S6C). In particular, S78 and/or S82 act as a conformational switch which plays an important role in regulating HSPB1 dimer formation and promotes chaperone activity (Choi et al, 2019). Taken together, these findings demonstrate that TPP provides further molecular insight into pathways previously implicated in SARS-CoV-2 infection, as well as implicating novel pathways in SARS-CoV-2 infection.

### The antiviral effects of heat shock chaperone and cytochrome P450 inhibitors

We investigated whether thermal stability or abundance changes can reveal functionally relevant host proteins that could be exploited as potential drug targets to block SARS-CoV-2 replication. Based solely on our ability to find compounds that selectively modulate the activity of host proteins which were significantly altered in our data, we manually shortlisted seven druggable host target proteins which we found to exhibit altered abundance (CYP1A1: inhibited by rhapontigenin) or thermostability (CAPN1: PD-150606, IMPDH1/2: ribavirin and mycophenolic acid, HSP90: tanespimycin, PTRG1: acetylsalicylic acid and diclofenac, CTSV: E64d and CA-074-Me) upon SARS-CoV-2 infection. To account for effects of the compounds on both viral entry and replication, cells were pretreated with compounds and these were kept present throughout the experiment. The capacity of the tested compounds to inhibit SARS-CoV-2 proliferation was assessed by quantifying the amount double-stranded viral RNA after 20–24 h postinfection and their effects on cell viability were tested in parallel on uninfected cells. Two of the tested compounds suppressed SARS-CoV-2 proliferation (Fig 5) (discussed below). The other compounds showed no effect

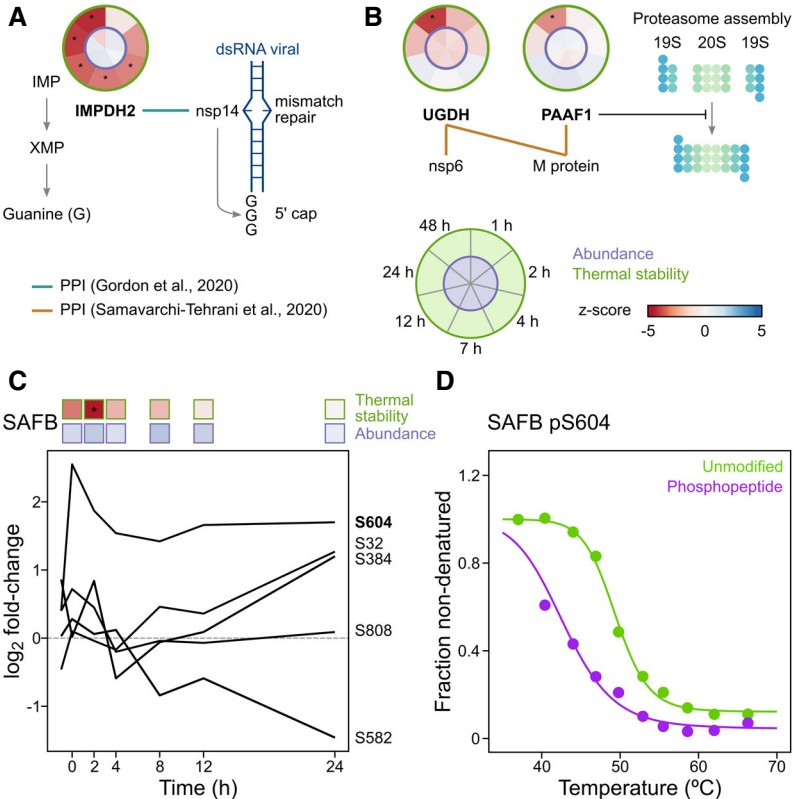

**Figure 4. Thermal stability changes in SARS-CoV-2-targeted host proteins.**

A  IMPDH2 was consistently destabilized from 4 h post-SARS-CoV-2 infection. IMPDH2 was previously shown to form a protein–protein interaction with nsp14 (green line) (Gordon *et al*, 2020b) and be a crucial factor in supporting SARS-CoV-2 replication (Gordon *et al*, 2020a). The SARS-CoV-1 nsp14 has a dual role in base pair mismatch repair and synthesis of the Guanine-rich 5′ cap of viral RNA (Chen *et al*, 2009; Smith & Denison, 2013; Smith *et al*, 2015). Circular thermal stability and abundance plots are from Caco-2 cells infected with SARS-CoV-2 as described in Fig 1A (see also Fig 1A for plot legend). Asterisk denotes significant regulation (see Materials and Methods).

B  Thermal destabilization of UGDH and PAAF1 temporally coincide and both were previously shown to interact with the SARS-CoV-2 proteins nsp6 (UGDH), and M (UGDH and PAAF1) (preprint: Samavarchi-Tehrani *et al*, 2020). PAAF1 negatively regulates the proteasome by controlling its assembly/disassembly (Park *et al*, 2005) and has been previously linked to transcription of HIV-1 (Lassot *et al*, 2007; Nakamura *et al*, 2012). Significant protein changes (abundance and thermal stability) are represented as circle plots for individual proteins, as previously described (Becher *et al*, 2018). Inner circle corresponds to protein abundance and the outer circle corresponds to thermal stability changes (*n* = 3). Asterisk denotes significant regulation (see Materials and Methods).

C  SAFB phosphopeptide abundance changes during SARS-CoV-2 infection. Vero cell phosphopeptide data from (Bouhaddou *et al*, 2020). Data previously acquired from Vero cells infected with SARS-CoV-2 (Bouhaddou *et al*, 2020). S604 phosphosite that temporally coincides with SAFB thermal destabilization (preprint: Potel *et al*, 2020). Shown above the line plot are the thermostability and abundance profiles of SAFB in SARS-CoV-2 infected Caco-2 cells as described in Fig 1A (*n* = 3). Asterisk denotes significant regulation (see Materials and Methods).

D  Thermal stability profile for all unmodified SAFB peptides (green) and the phosphorylated S604 (pS604) (purple) in HeLa cells. Data from Potel *et al* (preprint: Potel *et al*, 2020).

on SARS-CoV-2 proliferation in the concentration range tested (Appendix Fig S7A–H), this does not rule out that they could be active in combination with other drugs or in different cellular systems, as is the case for ribavirin (Bojkova *et al*, 2020) and the CAPN1 inhibitor (Ma *et al*, 2020).

The overall broad and consistent thermal destabilization of chaperones involved in unfolded protein folding suggested a global shift in their engagement with unfolded client proteins (Fig 2B). This could be due to the drastic increase in chaperone occupancy with large volumes of unfolded viral protein substrates, which may translate to a detectable shift in chaperone thermal stability. We verified the functional relevance of HSP90AA1 and HSP90AB1 thermal destabilization (Fig 5A) using the selective inhibitor tanespimycin,

which efficiently suppressed viral replication at low micromolar concentrations ($EIC_{50} \sim 2\ \mu M$) that were not toxic to the human cells (Fig 5B). In addition, we detected a pronounced increase in CYP1A1 abundance between 4 and 12 h postinfection, which was then followed by a strong decrease between 24 and 48 h postinfection (Fig 5C). Interestingly, treatment with the selective CYP1A1 inhibitor, rhapontigenin, suppressed SARS-CoV-2 proliferation within the micromolar range ($EIC_{50} \sim 50\ \mu M$) (Fig 5D).

Taken together, these results demonstrate that changes in protein abundance or thermal stability can suggest molecular targets for viral replication inhibition. This shows the utility of TPP for uncovering infection-relevant host pathways, which can expedite the identification of potential antiviral therapies.

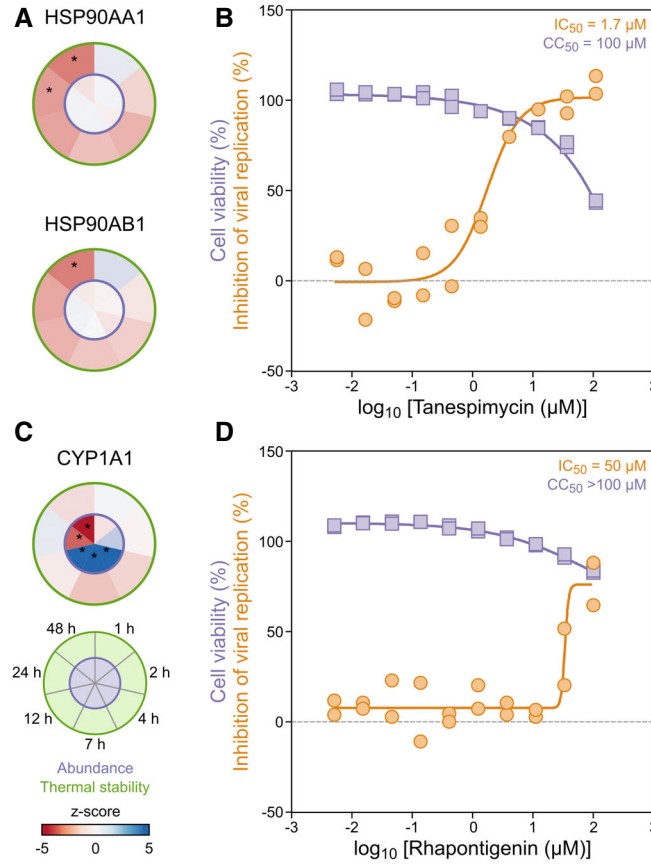

**Figure 5. 2D-TPP abundance and stability changes uncover functionally relevant host targets during SARS-CoV-2 infection.**

A  Heat shock proteins, HSP90AA1 and HSP90AB1, were thermally destabilized during SARS-CoV-2 infection of Caco-2 cells (circular plots, left side (*n* = 3); see Fig 1A for circular plot legend). Significant protein changes (abundance and thermal stability) are represented as circle plots for individual proteins, as previously described (Becher *et al*, 2018). Inner circle corresponds to protein abundance and the outer circle corresponds to thermal stability changes (*n* = 3). Asterisk denotes significant regulation (see Materials and Methods).

B  The corresponding HSP90 inhibitor, Tanespimycin, effectively suppressed SARS-CoV-2 proliferation (*n* = 2). Calu-3 cells were pre-incubated with the compound, followed by the addition of SARS-CoV-2 and subsequent co-incubation in the presence of the compound for 20–24 h. Viral load was quantified in parallel using antibody-based detection of double-stranded RNA (orange). Compound toxicity in Calu-3 cells was assessed (purple) treated with the compound alone for 20–24 h.

C  The aryl hydrocarbon hydroxylase (CYP1A1) increased in abundance between 4 and 12 h postinfection and was then reduced between 24 and 48 h postinfection (circular plot, left, (*n* = 3); see Fig 1A for circular plot legend). Circular plots displayed as described in panel A. Asterisk denotes significant regulation (see Materials and Methods).

D  The CYP1A1 inhibitor suppressed SARS-CoV-2 proliferation (*n* = 2). Dose–response curves were performed as described in Fig 4A. See Appendix Fig S7 for additional tested drugs. Data displayed as described in (B).

## Discussion

This work describes a modified TPP workflow that can be used to probe the infection interface of diverse human pathogens requiring high biosafety containment. Here, we assessed for the first time

temporal alterations in host protein thermal stability upon SARS-CoV-2 infection. We observed pronounced shifts in host protein thermal stability across diverse host cellular processes and throughout distinct infection phases, e.g., mRNA splicing machinery (early) and protein folding/chaperones (mid to late). We then demonstrated that proteins exhibiting changes in thermal stability or abundance can highlight functionally relevant host targets that could be exploited as possible antiviral therapies. It is, however, important to note that changes in protein abundance or thermal stability may reflect a cellular defense response, for which their pharmacological interference may instead enhance viral replication. Although none of the tested compounds led to an increase in viral load, such effects may occur.

We sought to interfere with SARS-CoV-2-induced HSP90 destabilization by treatment with the HSP90 selective inhibitor tanespimycin. This resulted in efficient inhibition of SARS-CoV-2 proliferation, providing compelling evidence for the requirement of HSP90 proteins in SARS-CoV-2 proliferation. This corroborates recent findings whereby HSP90AA1 mRNA transcripts were found to be upregulated in a lung carcinoma cell line (H1299) infected with SARS-CoV-2 (preprint: Emanuel *et al*, 2020)—prompting the authors to reveal that HSP90AA1 inhibition with tanespimycin similarly blocked SARS-CoV-2 replication in Calu-3 cells. Due to the non-specific nature of HSP90 client proteins, their inhibition should be effective against diverse viruses. Indeed, HSP90 is considered to be universally required for viral protein homeostasis (Geller *et al*, 2012). For instance, tanespimycin, or the related HSP90 inhibitor geldanamycin, effectively suppress diverse viral pathogens such as the Mumps virus and porcine reproductive and respiratory syndrome virus *in vitro* (Gao *et al*, 2014; Katoh *et al*, 2017). Furthermore, tanespimycin is undergoing clinical trials for cancer treatment and has been proposed as an effective broad-spectrum antiviral drug (Geller *et al*, 2012; Wang *et al*, 2017). Thus, HSP90 inhibition warrants further exploration as a broad-spectrum host-directed therapeutic against SARS-CoV-2.

One of the many pathways regulated by HSP90 chaperones includes the aryl hydrocarbon receptor (AhR) signalling pathway (Pongratz *et al*, 1992). During SARS-CoV-2 infection, AhR promotes interferon-induced mucus production in a rodent model (Liu *et al*, 2020). This build-up of alveolar mucus is thought to underly the obstructed blood–gas exchange seen in SARS-CoV-2 patients, thus leading to hypoxia and reduced lung capacity. In line with a role for AhR activation during SARS-CoV-2 infection, we detected a pronounced increase in the abundance of the well-known target gene product, CYP1A1 (4–12 h postinfection). Interestingly, pharmacological inhibition of HSP90 with tanespimycin is known to potently suppress CYP1A1 abundance by promoting proteasome-mediated degradation of the upstream transcriptional regulator, AhR (Hughes *et al*, 2008; Kasai & Kikuchi, 2010). It is therefore conceivable that at least part of the mechanism by which tanespimycin suppresses SARS-CoV-2 replication is through reducing CYP1A1 abundance, whose activity we show is required for SARS-CoV-2 replication. Although it remains unclear how CYP1A1 activity is required for SARS-CoV-2 replication in this context, further exploration of HSP90-dependent pathways required for SARS-CoV-2 replication could lead to the identification of additional therapeutic host targets.

In conclusion, we assessed biophysical changes in host proteins induced by a major human pathogen, SARS-CoV-2. We anticipate

this resource will aid in understanding of SARS-CoV-2 pathogenesis mechanisms and assist with rational design of host-targeted therapeutics that block SARS-CoV-2 infection. More broadly, applying TPP to organisms requiring high biosafety containment will unveil a hidden dimension of molecular interplay between highly infectious pathogens and their host.

# Materials and Methods

### Chemicals

The following compounds were purchased from Sigma; PD-150606 (D5946), ribavirin (R9644), mycophenolic acid (M3536), tanespimycin (A8476), acetylsalicylic acid (A5376), diclofenac (D689), rhapontigenin (PHL83903) and E64d (E8640). The cathepsin inhibitor CA-074me (205531) was purchased from Merck Millipore. Remdesivir was purchased from MedChemExpress (HY-104077). Drug stocks were prepared as follows: PD-160606 (50 mM in DMSO), ribavirin (50 mM in $H_2O$), mycophenolic acid (100 mM in MeOH), tanespimycin (50 mM in DMSO), acetylsalicylic acid (100 mM in EtOH), diclofenac (10 mM in MeOH), rhapontigenin (10 mM in DMSO), CA-074me (20 mM in DMSO), E64d (10 mM in DMSO) and Remdesivir (10 mM in DMSO).

### Cell culture and virus stock

Caco-2 cells were acquired from APC Cork (Alimentary Pharmabiotic Centre, University College Cork, National University of Ireland, Cork, Ireland), HeLa cells (CCL-2), VeroE6 and Calu-3 cells from ATCC. All cells below passage 35 were used for experiments. Caco-2 cells were propagated in DMEM with GlutaMAX Gibco Cat No 61965-026 and 20% heat inactivated foetal bovine serum (FBS). Vero cell culture and virus particle generation and titration were performed as previously described (Stanifer *et al*, 2020). VeroE6 (African green monkey kidney epithelial cell line) and Calu-3 (human lung epithelial cell line) cells were cultured in Dulbecco's modified Eagle medium (DMEM, Life Technologies) containing 10% or 20% foetal bovine serum, respectively, 100 U/ml penicillin, 100 μg/ml streptomycin and 1% non-essential amino acids (complete medium). Cell lines used in these experiments tested negative for mycoplasma using the MycoAlert Plus mycoplasma detection kit (Lonza) as per manufacturer's instructions.

### Visualization of SARS-CoV-2 viral N-protein during infection

Caco-2 cells were seeded on iBIDI glass bottom 8-well chamber slides. At indicated times postinfection, cells were fixed in 4% paraformaldehyde (PFA) for 20 min at room temperature. Cells were washed and permeabilized in 0.5% Triton X-100 for 15 min at room temperature. Mouse monoclonal antibody against SARS-CoV NP (Sino biologicals MM05) were diluted in phosphate-buffered saline (PBS) at 1:1,000 dilution and incubated for 1 h at room temperature. Cells were washed in 1× PBS three times and incubated with Goat anti-mouse Alexa Fluor 568 secondary antibody and DAPI for 45 min at room temperature. Cells were washed in 1× PBS three times and maintained in PBS. Cells were imaged by epifluorescence on a Nikon Eclipse Ti-S (Nikon).

### 2D-TPP infection time course

Two days prior to infection, $1.5 \times 10^6$ Caco-2 cells were seeded in triplicate into plastic 10 cm tissue culture dishes (Greiner Cell State) per condition or in 8-well iBIDI chambers for immunofluorescence staining. Caco-2 cells were infected with SARS-CoV-2 (BetaCoV/Germany/BavPat1/2020 p.1) at an MOI of 0.5 for 1 h. Following 1 h infection, virus supernatants were removed and cells were washed, and fresh media was added to cells At indicated times, the 10 cm dish samples were harvested as described below (see thermal proteome profiling) and the iBIDI chambers were fixed and stained as described above (visualization of SARS-CoV-2). The percentage of infected cells was determined by creating a nuclei mask in Ilastik (www.ilastik.org). CellProfiler (www.cellprofiler.org) was then used to measure the fluorescence intensity inside each nucleus of the mask. The values were thresholded using mock cells as a background control, and the percentage of infected cells was calculated by the ratio of nuclei to positive cells. Visual inspection of infected cells revealed the appearance of cytopathogenic effects from 48 h postinfection.

### Thermal proteome profiling

Thermal proteome profiling was performed as previously described (Becher *et al*, 2018) with the following modifications. The infected Caco-2 cells were harvested at 1, 2, 4, 7, 12, 24 and 48 h postinfection. For harvesting, cells were trypsinized, washed once with PBS and resuspended in 220 μl PBS. Each 20 μl of the concentrated cells were pipetted into a 96-well PCR plate and the samples from each time point were subjected to a thermal gradient (40.0, 42.1, 43.8, 46.5, 50.0, 54.0, 57.3, 60.1, 62.0, 64.0°C) for 3 min in a thermocycler (MJ Research, PTC-0200 DNA Engine) followed by 3 min at room temperature. The cells were then placed on ice and lysed with 30 μl lysis buffer (0.8% NP-40, 1 mM $MgCl_2$, 1× protease inhibitor (Roche), 1× phosphatase inhibitor (PhosStop, Sigma-Aldrich), 250 U/ml benzonase in PBS) for 1 h, shaking at 4°C and 500 rpm. A 0.45 μm 96-well filter plate (Millipore, ref: MSHVN4550) was pre-wet with 50 μl of 0.8% NP-40 in PBS by centrifugation, an additional 100 μl of 0.8% NP-40 in PBS was added to each sample, and the samples were transferred to the pre-wet filter plate. The filter plate was transferred to an extraction plate vacuum manifold for Oasis 96-well plates from Waters (Cat. 186001831), and the sample was filtered for the removal of protein aggregates. To verify the effect of the heat treatment, the soluble protein concentration at each temperature for each experiment was determined using the BCA assay, according to the manufacturer's instructions (Thermo Fisher Scientific). Then, 100 μl of each sample was transferred to a new PCR plate, 10 μl of denaturing buffer (20 mM TCEP (tris (2-carboxyethyl)phosphine) in 2% SDS) was added, and the plate was covered with aluminium foil, boiled for 10 min at 95°C and kept at −20°C until prepared for mass spectrometry.

### Mass spectrometry-based proteomics

Proteins were digested according to a modified SP3 protocol (Hughes *et al*, 2014, 2019). Briefly, approximately 5 μg of protein was diluted in 20 μl of water and added to the bead suspension (10 μg of beads (Thermo Fischer Scientific—Sera-Mag Speed Beads, (4515-2105-050250 and 6515-2105-050250) in 10 μl 15% formic acid and 30 μl

ethanol). After a 15-min incubation at room temperature with shaking, beads were washed four times with 70% ethanol. Next, proteins were digested overnight by adding 40 μl of digest solution (5 mM chloroacetamide, 1.25 mM TCEP, 200 ng trypsin and 200 ng LysC in 100 mM HEPES pH 8). Peptides were then eluted from the beads, dried under vacuum, reconstituted in 10 μl of water and labelled for 30 min at room temperature with 45 μg of TMTpro (Thermo Fisher Scientific) dissolved in 4 μl of acetonitrile. The reaction was quenched with 4 μl of 5% hydroxylamine for 15 min at room temperature, and experiments belonging to the same mass spectrometry run were combined. Samples were desalted with solid-phase extraction by loading the samples onto a Waters OASIS HLB μElution Plate (30 μm), washing them twice with 100 μl of 0.05% formic acid, eluting them with 100 μl of 80% acetonitrile and 0.05% formic acid, and drying them under vacuum. Finally, samples were fractionated onto 12 fractions on a reversed-phase C18 system running under high pH conditions. This consisted of an 85 min gradient (mobile phase A: 20 mM ammonium formate (pH 10) and mobile phase B: acetonitrile) at a 0.1 ml/min starting at 0% B, followed by a linear increase to 35% B from 2 to 60 min, with a subsequent increase to 85% B from up to 62 min and holding this up to 68 min, which was followed by a linear decrease to 0% B up to 70 min, finishing with a hold at this level until the end of the run. Fractions were collected every 2 min from 12 to 70 min and every 12[th] fraction was pooled together.

Samples were analysed with liquid chromatography coupled to tandem mass spectrometry, as previously described. Briefly, peptides were separated using an UltiMate 3000 RSLCnano system (Thermo Fisher Scientific) equipped with a trapping cartridge (Precolumn; C18 PepMap 100, 5 μm, 300 μm i.d. × 5 mm, 100 Å) and an analytical column (Waters nanoEase HSS C18 T3, 75 μm × 25 cm, 1.8 μm, 100 Å). Solvent A was 0.1% formic acid in LC-MS grade water, and solvent B was 0.1% formic acid in LC-MS grade acetonitrile. Peptides were loaded onto the trapping cartridge (30 μl/min of 0.05% trifluoroacetic acid in LC-MS grade water for 3 min) and eluted with a constant flow of 0.3 μl/min using a 120 min analysis time (with a 2–30% B elution, followed by an increase to 40% B, and a final wash to 80% B for 2 min before re-equilibration to initial conditions). The LC system was directly coupled to a Q Exactive Plus mass spectrometer (Thermo Fisher Scientific) using a Nanospray-Flex ion source and a Pico-Tip Emitter 360 μm OD × 20 μm ID; 10 μm tip (New Objective). The mass spectrometer was operated in positive ion mode with a spray voltage of 2.3 kV and capillary temperature of 275°C. Full-scan MS spectra with a mass range of 375–1,200 $m/z$ were acquired in profile mode using a resolution of 70,000 (maximum fill time of 250 ms or a maximum of 3e6 ions (automatic gain control, AGC)). Fragmentation was triggered for the top 10 peaks with charge 2–4 on the MS scan (data-dependent acquisition) with a 30-s dynamic exclusion window (normalized collision energy was 30), and MS/MS spectra were acquired in profile mode with a resolution of 35,000 (maximum fill time of 120 ms or an AGC target of 2e5 ions).

## Protein identification and quantification

MS data were processed as previously described (Franken et al, 2015). Briefly, raw MS files were processed with isobarQuant (Franken et al, 2015), and the identification of peptides and proteins was performed with Mascot 2.4 (Matrix Science) against the human

(Proteome ID: UP000005640) and SARS-CoV-2 (Proteome ID: UP000464024) UniProt FASTA, modified to include known contaminants and the reversed protein sequences (search parameters: trypsin; missed cleavages 3; peptide tolerance 10 ppm; MS/MS tolerance 0.02 Da; fixed modifications were carbamidomethyl on cysteines and TMTpro on lysine; variable modifications included acetylation on protein N-terminus, oxidation of methionine and TMTpro on peptide N-termini).

## Abundance and thermal stability score calculation

We calculated abundance and thermal stability scores for every protein at every infection time point by combining the data from the three replicates similarly to previously described (Mateus et al, 2018). Briefly, the overall distribution of signal sum intensities was normalized with vsn to compensate for slight differences in protein amounts from each TMT channel. Then, for every protein, we calculated the ratio of the signal sum intensity of each time point to the signal sum of the uninfected sample at the same temperature. The abundance score of each protein at each time point was calculated as the average $\log_2$ fold change at the two lowest temperatures weighted for the number of temperatures in which the protein was identified for each replicate (requiring that there was data for two biological replicates). The thermal stability score of each protein at each time point was then calculated by subtracting the abundance score from the log2 fold changes of all temperatures, and summing the resulting fold changes weighted for the number of temperatures in which the protein was identified for each replicate (requiring that there were at least ten data points to calculate this score). To assess the significance of abundance and thermal stability scores, we used a limma analysis, followed by an FDR analysis, using the fdrtool package (see analysis script at: https://github.com/andrenmateus/TPP-SARSCoV2/). Abundance and thermal stability scores for all time points were separately transformed to $z$-scores. Proteins with calculated $|z$-score$| > 1.96$ (corresponding to a global $P < 0.05$ for the effect size) and with $q$-value $< 0.05$ were considered significantly changed.

## Network analysis

All proteins having significant stability changes during infection with SARS-CoV-2 were selected, given the same weight and mapped into a custom made human interactome integrating STRING v 11.0 (Edge Score > 0.75) and Opentargets interactome (November 2019) (compilation of Intact, Reactome and Signor). All edges were treated as undirected, redundancies and self-loops removed, and a personalized PageRank algorithm was used to propagate the signal. Walktrap clustering was performed in the network regions with the highest page rank score (third quantile) to produce modules of interacting genes with no more than 300 genes. Modules were selected as significant if proteins with stability changes were enriched (Fisher test with BF multiple testing $P$ value adjustment) or if the distribution of page rank score was significantly higher (Kolmogorov–Smirnov test with BF multiple testing $P$ value adjustment). Finally, we selected the significant modules that were enriched in proteins with: SARS-CoV-2 viral interactors (Gordon et al, 2020b), phosphosites changing during SARS-CoV-2 infections or kinases whose activity is affected by SARS-CoV-2 infection (Bouhaddou et al, 2020). The resulting six modules were further simplified

keeping only the nodes that were changing at any level during infection (protein stability, phosphosite dynamics, kinase activity, PPI with viral proteins) and the edges connecting them.

## TPP data comparison with protein–protein interaction studies

For Gordon et al (2020a) we took the data from Table S2 (of the corresponding work), sheet "SARS-CoV-2_HighConfidence" and compared the overlap of the column PreyGene with our hits. For Samavarchi-Tehrani et al (preprint: Samavarchi-Tehrani et al, 2020): we took data from Table S2 (of the corresponding work), sheet "viral baits (N- & C-term merge)" filtered for BFDR < 0.01 and compared the overlap of the column PreyGene with our hits. For Bouhaddou et al (2020) we took data from Table S1 (of the corresponding work), sheet "PhosphoDataFiltered" filtered for any of the columns $\log_2 FC$ for $> 1$ or $< -1$ and in adj.Pvalue $< 0.05$ and compared the overlap of the column Gene_Name with our TPP hits.

## Antiviral activity test

For testing the antiviral activity and cytotoxicity of selected compounds, Calu-3 cells were seeded 1 day prior to infection at a density of $3 \times 10^4$ cells per well of a clear 96-well plate (Corning). Antiviral activity and cytotoxicity were tested in duplicates, and remdesivir was included as positive control in each test run (concentration range: 0.5 nM – 10 μM in 3-fold serial dilutions). For the cytotoxicity test, cells were left untreated or treated with the respective drug in 3-fold dilution steps (for concentration ranges, see Fig 5 and Appendix Fig S6). For the antiviral activity test, cells were additionally infected with SARS-CoV-2 at an MOI of 1. Plates were incubated at 37°C for 20–24 h. Cytotoxicity was measured using a CellTiter-Glo Luminescent Cell Viability Assay (Promega) according to the manufacturer's instruction. For the antiviral activity test, plates were fixed with 6% formaldehyde, washed with PBS and applied to immunostaining using a double-strand RNA-specific antibody (Scicons) suitable to detect SARS-CoV-2 infected cells. Cells were permeabilized with 0.2% Triton X-100 in PBS for 15 min at room temperature, washed once with PBS and blocked in 2% milk in PBS with 0.02% Tween-20 for 1 h at room temperature. Cells were incubated with the mouse anti-double-strand RNA antibody for 1 h at room temperature, washed three times with PBS, incubated with the secondary antibody (anti-mouse IgG, conjugated with horseradish peroxidase) for 1 h at room temperature and subsequently washed four times with PBS. Detection was done by adding TMB (3,3′,5,5′-Tetramethylbenzidine) substrate (Thermo Fisher Scientific), and plates were analysed by photometry at 620 nm using a plate reader. The background absorbance was measured at 450 nm. The percentage of cell viability and inhibition was determined by dividing the values obtained from the drug-treated cells by the values from the untreated controls. $IC_{50}$ values were calculated by non-linear regression sigmoidal dose–response analysis using the GraphPad Prism 7 software package.

## Data availability

The mass spectrometry proteomics data have been deposited to the ProteomeXchange Consortium via the PRIDE partner repository with the dataset identifier PXD022115 (https://www.ebi.ac.uk/pride/archive/projects/PXD022115).

All code has been deposited at https://github.com/andrenmateus/TPP-SARSCoV2/.

**Expanded View** for this article is available online.

## Acknowledgements

We acknowledge funding from EMBL for this research. AM, KM, CGPV and CMP were supported by a fellowship from the EMBL Interdisciplinary Postdoc (EI3POD) Programme under Marie Skłodowska-Curie Actions COFUND (grant number 664726). MS and SB were supported by research grants from the Deutsche Forschungsgemeinschaft (DFG): (Project number 240245660, 278001972, 415089553 and 272983813 to SB and 416072091 to MS), the state of Baden-Wuerttemberg (AZ: 33.7533.-6-21/5/1) and the Bundesministerium für Bildung und Forschung (BMBF, grant number 01KI20198A); RB by the German Center for Infection Research (DZIF), project numbers 8029801806 and 8029705705; and AT by an ERC consolidator grant, uCARE (ID 819454). We thank the EMBL Genomics Core Facility for providing a thermal cycler for use in the BSL3 lab. Open Access funding enabled and organized by Projekt DEAL.

## Author contributions

Investigation: JS, MS, AM, KM, MR, HK, CGPV, PW, CK, AJ and CP. Data analysis: MS, AM, IB-H, NK and FS Writing: All authors contributed to writing of the manuscript. Figures: AM and IB-H. Supervision: MMS, AT, PB, MS, SB, RB and BK. Funding acquisition: MMS, AT, PB, MS, SB and RB.

## Conflict of interest

BK is a founder and shareholder of OmicScouts and MSAID. He has no operational role in either of the two companies. All other co-authors declare no competing financial interests.

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
