## [Review Process File · Molecular Systems Biology]

SARS-CoV-2 infection remodels the host protein thermal stability landscape

Joel Selkrig, Megan Stanifer, André Mateus, Karin Mitosch, Inigo Barrio-Hernandez, Mandy Rettel, Heeyoung Kim, Carlos Voogdt, Philipp Walch, Carmon Kee, Nils Kurzawa, Frank Stein, Clément Potel, Anna Jarzab, Bernhard Küster, Ralf Bartenschlager, Steeve Boulant, Pedro Beltrao, Athanasios Typas, and Mikhail Savitski

DOI: 10.15252/msb.202010188

Corresponding author(s): Mikhail Savitski (mikhail.savitski@embl.de), Athanasios Typas (athanasios.typas@embl.de), Pedro Beltrao (pbeltrao@ebi.ac.uk), Steeve Boulant (s.boulant@dkfz-heidelberg.de)

Review Timeline:

Submission Date:	21st Dec 20
Editorial Decision:	22nd Dec 20
Revision Received:	11th Jan 21
Editorial Decision:	14th Jan 21
Revision Received:	15th Jan 21
Accepted:	18th Jan 21

Editor: Maria Polychronidou

Transaction Report:

The reviewers' comments and authors' responses are not available with this article, as the initial review process took place with another journal.

Thank you again for sending us your manuscript "SARS-CoV-2 infection remodels the host protein thermal stability landscape". The reviews from the other journal are quite constructive and we can definitely use them rather than reviewing the study from scratch. I would therefore invite you to submit your manuscript once you have performed the revisions addressing the issues raised. Please include a detailed point by point response, so that we can easily evaluate the changes.

In summary, the two main issues raised were the lack of a decisive methodological advance and the relatively limited biological insights into SARS-CoV-2 infection. The methodological advance is not an issue for Molecular Systems Biology, as we do not see this as a Method paper. In my view, and along the lines of the reviewers' comments, the value of the study is mostly at the resource level. Regarding the level of biological insight, we do not think that extensive functional follow up analyses or deep novel mechanistic insights are essential for acceptance in Molecular Systems Biology. That said, I think that it would be important to provide some further support for the value of the presented resource (and its combination with the PPI and phosphorylation data from previous work) for understanding SARS-CoV-2 infection and identifying drug targets.

As you have not yet provided your revision plan, I am not sure what additional data you have at hand, but looking at the reviewers' comments I would propose the following:

- While a fully fletched functional follow up on CYP1A1 does not seem necessary, in case you have some further data it would be nice to strengthen this part of the study and increase the overall level of biological insight, which was a general concern raised by the reviewers.
- Additional assessment of viral replication by qPCR of the supernatant as independent validation of the compound effects would be a nice addition.
- In order to increase the confidence in the value of the resource, some evidence that the prioritized drugs could not have been identified by other approaches would need to be provided.
- Comparing protein destabilization and stabilization seems interesting and relatively easily addressable.

We thank you for considering our manuscript and appreciate the opportunity to resubmit our revised version according to the reviewers' comments from the other journal. Please find our detailed point by point response to the reviewers' comments below as well as our response to the points you raised in our previous correspondence on December 22nd 2020.

Due to the current Germany-wide lockdown regulations which limit experimental work, our revisions focus primarily on addressing the comments raised by rewriting sections of the text and expanding on some of the issues raised to more clearly illustrate the biological insight provided by our findings.

1. While a fully fletched functional follow up on CYP1A1 does not seem necessary, in case you have some further data it would be nice to strengthen this part of the study and increase the overall level of biological insight, which was a general concern raised by the reviewers.

We have not acquired additional data on CYP1A1 however, we have revised the discussion to further highlight the biological significance of CYP1A1 in SARS-CoV-2 infection, particularly with respect to upstream regulators i.e. AhR and HSP90. We explain how HSP90 inhibition – which suppresses CYP1A1 expression in Caco-2 cells – serves as indirect evidence of the biological relevance of CYP1A1 for SARS-CoV-2 infection.

L306-320: "One of the many pathways regulated by HSP90 chaperones includes the aryl hydrocarbon receptor (AhR) signaling pathway (Pongratz, 1992). During SARS-CoV-2 infection, AhR promotes interferon-induced mucus production in a rodent model (Liu et al., Cell Research). This buildup of alveolar mucus is thought to underly the obstructed blood-gas exchange seen in SARS-CoV-2 patients, thus leading to hypoxia and reduced lung capacity. In line with a role for AhR activation during SARS-CoV-2 infection, we detected a pronounced increase in the abundance of the well-known target gene product, CYP1A1 (4-12 hours post infection). Interestingly, pharmacological inhibition of HSP90 with 17-AAG is known to potently suppresses CYP1A1 abundance by promoting proteasome-mediated degradation of the upstream transcriptional regulator, AhR (Kasai et al, 2010; Hughes et al., 2008). It is therefore conceivable that at least part of the mechanism by which 17-AAG suppresses SARS-CoV-2 replication is through reducing CYP1A1 abundance, whose activity we show is required for SARS-Cov-2 replication. Although it remains unclear how CYP1A1 activity is required for SARS-CoV-2 replication in this context, further exploration of HSP90 dependent pathways required for SARS-CoV-2 replication could lead to the identification of additional therapeutic host targets."

2. Additional assessment of viral replication by qPCR of the supernatant as independent validation of the compound effects would be a nice addition.

We agree that independent validation by qPCR of culture supernatants would provide additional confirmation of the effects of these compounds on SARS-CoV-2 replication. However, these were not collected at the time of the previous experiments and these experiments would need to be repeated. Given the current lockdown, and the uncertainty of its duration, such experiments are unfeasible for us at this point in time.

3. In order to increase the confidence in the value of the resource, some evidence that the prioritized drugs could not have been identified by other approaches would need to be provided.

We prioritized a unique panel of ten drugs solely based on their selectivity for seven distinct host targets that displayed thermal stability (6/7) or abundance (1/7) changes upon SARS-CoV-2 infection entirely based on our data. We note that four of the seven prioritized drugs have been previously tested by others for SARS-CoV-2 inhibition based on indirect evidence (e.g. overall pathway enrichments) and not due to direct detection of changes in the protein state of the drug target itself. We have elaborated on this in our response to reviewer # 2 point #3 below and have amended the main text to clarify how we prioritized drugs as well as the discussion to highlight the above point.

From: “We manually shortlisted seven druggable host target proteins which we found to exhibit altered abundance (CYP1A1: inhibited by rhapontigenin) or thermostability (CAPN1: PD-150606, IMPDH1/2: ribavirin and mycophenolic acid, HSP90: tanespimycin, PTRG1: acetylsalicylic acid and diclofenac, CTSV: E64d and CA-074-Me) upon SARS-CoV-2 infection.”

To L244-245 “*Based solely on our ability to find compounds that selectively modulate the activity of host proteins which were significantly altered in our data, we manually shortlisted seven druggable host target proteins which we found to exhibit altered abundance (CYP1A1: inhibited by rhapontigenin) or thermostability (CAPN1: PD-150606, IMPDH1/2: ribavirin and mycophenolic acid, HSP90: tanespimycin, PTRG1: acetylsalicylic acid and diclofenac, CTSV: E64d and CA-074-Me) upon SARS-CoV-2 infection.*”

4. Comparing protein destabilization and stabilization seems interesting and relatively easily addressable.”

We agree that this is indeed an interesting analysis to perform. We have now separated GO term enrichments based on the directionality of stability changes. This analysis is now included as supplementary Figure S3, S4 and referred to in the main text as follows;

L138-139: “Hits related to protein folding were mostly the result of thermal destabilization and cell-cell adhesion proteins were mostly stabilized (Figure S3, S4 and Table 2).”

Please also see our detailed response below.

****The remainder of the authors' point-by-point response to the reviewers' comments has been removed, as it contains the reviewers' comments from the initial review process, which took place with another journal.****

Thank you again for submitting your work to Molecular Systems Biology. As we discussed previously, the reviews from the other journal were quite constructive so we decided to use these reports rather than reviewing the study from scratch. Thank you for the detailed point by point response describing how the issues raised have been addressed. I feel that the additional analyses, discussions, edits and responses to the reviewers' comments seem to satisfactorily address the points raised. As most of the reviewers' concerns referred to the need to better support the main conclusions, but there were no concerns related to the technical aspects of the core experiments reported in the study, we have decided to proceed with making a decision based on our evaluation of the study and your responses, without involving an Editorial Advisory Board member or external expert.

We understand that given the current lockdown situation it is difficult to perform experiments. Therefore, we think that further experimental data on CYP1A1 are not required at this stage. Given that as the reviewers acknowledged, the main value of the study is at the resource level and considering that the study deals with a very timely and intensely researched topic, we have decided to proceed with publishing the study in its current form.

Before we can formally accept your study for publication in Molecular Systems Biology, I would ask you to perform some minor revisions listed below.

The authors have made all requested editorial changes.

Thank you again for sending us your revised manuscript. We are now satisfied with the modifications made and I am pleased to inform you that your paper has been accepted for publication.

Corresponding Author Name: Mikhail M. Savitski, Athanasios Typas, Pedro Beltrao, Steeve Boulant

Manuscript Number: MSB-2020-10188